# Fish and Black Soldier Fly Meals as Partial Replacements for Soybean Meal Can Affect Sustainability of Productive Performance, Blood Constituents, Gut Microbiota, and Nutrient Excretion of Broiler Chickens

**DOI:** 10.3390/ani13172759

**Published:** 2023-08-30

**Authors:** Youssef A. Attia, Fulvia Bovera, Khalid A. Asiry, Shatha Alqurashi, Majed S. Alrefaei

**Affiliations:** 1Sustainable Agriculture Production Research Group, Agriculture Department, Faculty of Environmental Science, King Abdulaziz University, Jeddah 21589, Saudi Arabia; kasiry@kau.edu.sa (K.A.A.); msalrefaei@stu.kau.edu.sa (M.S.A.); 2Sustainable Agriculture Production Research Group, Department of Veterinary Medicine and Animal Production, University of Napoli Federico II, Via F. Delpino 1, 80137 Napoli, Italy; 3Department of Biology, College of Science, University of Jeddah, Jeddah 23218, Saudi Arabia; saaqurshi@uj.edu.sa

**Keywords:** broilers, *Hermetia illucens*, supplementation, sustainability of natural resources, performance, gut microbiota

## Abstract

**Simple Summary:**

Alternative protein sources are essential due to the future perspectives of increasing the world’s human population. Insects could be a great opportunity to improve the sustainability of animal production. In this context, the black soldier fly is one of the most promising species, even if the best level of its inclusion in the broiler diet, as well as the best insect stage to use, is not completely defined. Our study showed that both larvae and prepupae meals had a positive effect on animal health, in particular on the intestinal microbiota.

**Abstract:**

One hundred and twenty, one-day-old male broiler chicks were used to investigate the effects of supplementation with different dietary protein sources on their performance and immune systems. Chicks were randomly divided into four equal experimental groups (six replicates, each of five chicks). The first group served as a control and was fed a standard corn–soybean meal diet. The second, third, and fourth groups were fed diets in which the soybean meal (SBM) was partly replaced by fish meal (FM), black soldier fly larvae (BSFL), and black soldier fly prepupae (BSFP), respectively. Throughout 1–14 and 15–42 days of age, FM, BSFL, and BSFP were added at 3 and 5%, respectively. The feed conversion ratio (FCR) of the FM group was the best among the tested groups. Feeding BSFP decreased final body weight (BW), BW gain, feed intake, and impaired the FCR compared to the other treatments for the entire experimental period. The BSFP group had significantly lower cecal *Salmonella* counts compared to the control group and lower total bacterial counts compared to the other groups except for BSFL. BSFL can be fed to broiler chickens at 3% during the starter period and 5% during the grower-finisher periods without negative influences on growth performance, red blood cell characteristics, blood lipid profiles, and nutrient excretion, while BSFP can improve the chickens’ gut ecosystem.

## 1. Introduction

Nowadays, alternative protein sources are essential due to the future perspectives of increasing the world’s human population [1] and limited grain supplies after the COVID-19 crisis and the Russia–Ukraine war. Insects are valuable protein supplies for animal nutrition [2,3,4] because of their high nutritive value and the limited natural resources (water, land, and feed) required for insect growth [5,6,7].

The primary and most expensive nutrient needed for broiler production is protein, and as chicken output rises, more protein is required to meet the demand for amino acids [8,9]. Recently, soybeans’ growing prices have also emerged as critical factors for the industry’s economic sustainability, particularly in some developing countries [10,11]. The most significant source of traditional animal protein in developing countries is fish meal, although there are substantial sustainability and pricing issues [12]. Fish meal has been applied extensively in chicken nutrition over the past few decades due to its high biological protein value and abundance of critical amino acids, particularly lysine and amino acids containing sulfur. In addition, fish meals can increase the omega-3 content in poultry products. However, the limitation in availability and the fluctuation of fish meal and soybean meal prices [13] entice the demand for alternative feed ingredients. Insect proteins can be a valuable alternative to soybean and fish meal [4], also considering that insects are normally included by poultry in their diet [14].

Several studies have shown that insect meals could replace soybean and fish meal in animal nutrition because of their low competitiveness with human food [15], high protein content [7,16], and low deleterious environmental impact [7,17,18].

The black soldier fly (BSF, *Hermetia illucens*, L., Diptera: Stratiomyidae) is the insect species with the greatest potential for industrial production. Black soldier fly larvae (BSFL) achieve a high growth rate and an excellent conversion of organic waste to complete their development [19], with a consistent amino acid concentration in their bodies when raised on distinct substrates [7,20,21]. It is generally recognized that using insects, especially BSFs, plays a crucial role in resolving problems relating to large amounts of organic waste dispersed around the Earth [4]. BSFs have been studied in chicken feeding lately as a partial or complete replacement for either soybean or fish meal [7,22,23,24]. Recently, Addeo et al. [11] indicated that low levels (3–5%) of insect meal inclusion in the diet of poultry could be considered a possible alternative approach due to its positive effects on animal production and health.

Thus, this work aimed to investigate the impact of fish and black soldier fly meals (from larvae, BSFL, and prepupae, BSFP) supplemented to diets as partial replacements for soybean meal on the productive performance sustainability, blood constituents, and gut microbiota of broiler chickens in comparison to a control corn–soybean-meal-based diet.

## 2. Materials and Methods

### 2.1. Diets

Fresh BSFL and BSFP were purchased from a local market (of commercial supply) in Saudi Arabia. The insect larvae and pupae were checked for general appearance and were in good form. BSFL and BSFP were dried in a forced hot-air oven at 65 °C until they reached a constant weight and ground to a fine powder to pass through a 0.5 mm sieve using an electric dry mill. The powder was then stored in well-tied black plastic bags at room temperature (≈25 °C). Samples of SBM, FM, BSFL, and BSFP were analyzed for chemical–nutritional characteristics (Table 1).

FM, BSFL, and BSFP were included in the formulations of the experimental diets biweekly and mixed well for ten minutes until the diets were totally homogenized. The ingredients and chemical–nutritional characteristics of the diets are reported in Table 2.

The chemical composition of the feed ingredients and the experimental diets was estimated in triplicate, according to the following AOAC [25] methods: 943.01 (DM), 920.39 (EE), 924.05 (ash), 954.01 (CP), 945.18 (CF), 973.18 (ADF), and 2002.04 (NDF) for dry matter (DM), ash, crude protein, ether extract (EE), and crude fiber (CF). The fiber was analyzed with a Velp fiber analyzer (FIWE 6) conforming to Van Soest et al. [26]. Chitin was determined in agreement with D’Hondt et al. [27]. The values of apparent metabolizable energy (AME) were calculated according to NRC [28].

### 2.2. Birds and Diets

All the animals were humanely treated according to the principles stated by the EC Directive 86/609/EEC (Council Directive, 2008) regarding the protection of animals used for experimental and other scientific purposes. The experimental procedures were approved by the Ethical Animal Care and Use Committee of King Abdulaziz University’s ethics code (ACUC-22-1-2). One hundred and twenty, one-day-old male commercial Arbor Acres broiler chicks with an average body weight of 42.8 g ± 1.5 were wing-banded and randomly divided into four experimental groups in a completely randomized design, with six replicates of five chicks each. Each replicate was kept in battery brooder wire cages (80 × 50 × 45 cm length–width–height). The control group was fed a diet without fish meal, BSFL, or BSFP inclusion. The 2nd, 3rd, and 4th groups were provided diets containing 3% fish meal, BSFL, and BSFP, respectively, as a partial replacement for soybean meal during the starter period. During the growing-finishing period, the SBM was replaced at 5% by FM, BSFL, and BSFP, respectively, for the 2nd, 3rd, and 4th groups. The inclusion levels of BSFL and BSFP were chosen based on the recent work by Addeo et al. [11]. The diets exploited in the starter period were administered from 1 to 14 days of age, whereas the diets of the growing-finishing period ranged from 15 to 42 days. All diets were formulated based on the Arbor Acres broiler management handbook [29]. Mash feed and water were provided ad libitum with a 23L:1D light–dark regime during the 1st week and a 20:4 light–dark cycle after that until the end of the experiment. The chickens were housed in a thermal control room (25 °C and 55% RH). The temperature inside the house was maintained at 33 °C for the first three days and then reduced by 3 °C each consecutive week until 25 °C. In the chicken chamber, the average minimum and maximum relative humidities were 53.2 and 64.5%, respectively. All birds were kept under the same managerial, hygienic, and environmental conditions. No coccidiostats, antibiotics, extra supplements, or acidifiers were employed in this research. The birds were not vaccinated, as the broiler house was being utilized for the 1st time.

At 14 and 42 days, body weight (BW) and feed intake (FI) were measured for each replicate and used to calculate body weight gain (BWG) and feed conversion ratio (FCR). The mortality rate was recorded daily. Excreta samples were collected (n = 12 per treatment as two per replicate) at 42 days of age. The excreta were cleaned from feathers, spared with perchloric acid, dried at 70 °C for 24 h, ground, and kept for further analyses. Chemical evaluation of excreta samples was determined as previously described for feed samples.

### 2.3. Hematological Characteristics

In each treatment group, two blood samples were obtained at the slaughter of 6 randomly selected birds at 42 days of age. The samples were taken in two tubes, with and without heparin. Blood was centrifuged at 1500× *g* for 20 min to obtain plasma and serum samples and frozen at −20 °C until being employed for analysis. 

Shortly after the collection, noncoagulated blood was applied for estimating red blood cell counts (RBCs) in line with Feldman et al. [30]. Hemoglobin (Hgb) concentrations and the percentage of packed cell volume (PCV %) were measured in agreement with Drew et al. [31]. The heterophils-to-lymphocytes ratio (H/L) was calculated by dividing the total count of heterophils by the total number of lymphocytes. 

Uric acid [32], plasma triglycerides (Trgs, mg/dL) [30], total plasma cholesterol (Chol, mg/dL) [33], and plasma HDL cholesterol [34] were determined. A high/low-density lipoprotein ratio was calculated. The risk of high cholesterol was attained by dividing the low-density lipoprotein by total cholesterol, conforming to Attia et al. [35]. The activity of serum aspartate aminotransferase (AST) and serum alanine aminotransferase (ALT) was measured, concurring with Ricard et al. [36].

### 2.4. Gut Microbial Counts

Cecum samples were collected for each treatment (n = 6) at the end of the experiment. The samples were diluted at 1:10 with normal saline, and approximately 100 mL of each dilution was plated on plate count agar (PCA), xylose lysine deoxycholate agar (XLD), eosin methylene blue agar (EMBA), tryptone glucose yeast extract agar (TGY), MacConkey agar, and reinforced clostridial agar (RCA) containing d-cyclo. All plates of PCA, XLD, EMB, TGY, and MacConkey agars were incubated at 37 °C for 24 h under aerobic conditions. Meanwhile, reinforced clostridial agar plates were incubated anaerobically at 37 °C for 24–48 h. The colonies’ counts of distinctive dilutions were recorded. The microbial counts were reported as colony-forming units (cfus) per ml of sample and converted into logs.

### 2.5. Statistical Analysis

The efficacy of sample size and the normal distribution of the data and their error were tested using the Shapiro–Wilk test [37]. The random distribution of the data collected for analyses confirmed the relevance of the 4 analysis of variance assumptions (ANOVA): (1) individual observation is mutually independent, (2) the data adhere to an additive-effect statistical model comprising fixed effects and random errors, (3) the random errors are normally distributed, and (4) the random errors have homogenous variations.

The homoscedasticity (variance homogeneity) was evaluated using Levene’s test SAS^®^ [37]. The data were statistically analyzed using one-way ANOVA of SAS^®^ [37], and the replicate was the experimental unit. The variables showing significant variations were tested using Tukey’s test [36]. The statistical model enrolled was as follows:Yij = μ + Ti + eij
where Yij = the dependent variable, μ = the overall mean, Ti = the effect of treatments, and eij = the random error.

## 3. Results

### 3.1. Chemical Composition of Feedstuffs

The proximate analysis (Table 1) exhibited that the crude protein content of the BSFL and BSFP samples (50.0 and 53.5% CP, respectively) was higher than that of SBM (46.2% CP). Moreover, BSFL and BSFP had higher ether extract content (22.0 and 20.0%, respectively) than yellow corn (6.50%), SBM (3.00%), and FM (9.69%). Chemical analysis revealed that the BSFL and BSFP samples contained 5.17 and 4.92% of chitin, respectively.

### 3.2. Growth Performance

No signs of diseases were observed in the chickens, which was confirmed by the absence of mortalities during the trial. The effects of the dissimilar diets on the growth of the broilers during 1–42 days of age are shown in Table 3. The BW at 1–14 days was not significantly contrasted among the experimental groups. However, the results highlighted that the protein supplements greatly affected the chicks’ final BW at 42 days of age and their BWG during days 15–42 and 1–42. The control, FM, and BSFL groups had significantly higher BWG from 15 to 42 days of age than the BSFP group. The BWG from 1 to 42 days of age and the final BW were statistically higher in the SBM, FM, and BSFL groups than in the BSFP group. Still, these groups did not statistically vary from the unsupplemented control group. 

The results of the feed intake from 1 to 42 days of age are displayed in Table 4. From 1 to 14 days of age, the broilers that were given a diet containing BSFL demonstrated a significantly greater consumption of feed compared to the other groups. Meanwhile, the BSFP group consumed the smallest amount of feed. From 15 to 42 days of age, groups that were fed FM and BSFL consumed more feed than the BSFP group. 

During 1–14 days of age, broilers fed BSFL showed an impaired FCR compared to the other groups. From 15 to 42 days, broilers fed BSFP exhibited the worst FCR, and the control group displayed the best one. For the whole experimental period, the FM group recorded the best FCR, but it was not statistically peculiar compared to the other groups, except for BSFP. The European Production Efficiency Index (EPEI) was similar for broilers that were fed the control diet, FM, and BSFL, while BSFP had the worst EPEI value.

### 3.3. Blood Profiles

The outcomes presented in Table 5 display that the discrete dietary protein source replacements (FM, BSFL, and BSFP) at 3 and 5% during the starter and growing-finishing periods, respectively, did not significantly impact the RBCs’ blood characteristics. RBCs, hemoglobin (Hgb), packed cell volume (PCV), mean corpuscular volume (MCV), mean corpuscular hemoglobin (MCH), and mean corpuscular hemoglobin concentration (MCHC) were statistically similar among the disparate groups.

The effects of supplementing disparate dietary protein sources (FM, BSFL, and BSFP) on serum lipid profiles are illustrated in Table 6. Replacement of SBM with FM, BSFL, and BSFP at 3 and 5% during the starter and growing-finishing periods, respectively, did not significantly influence the serum lipid profile, including total plasma lipid (TL), triglycerides (Trgs), total cholesterol (Chol), high- and low-density lipoproteins (LDLs) and their ratio, and very low-density lipoprotein (VLDL). Nevertheless, the differences in high cholesterol risk approached significance (*p* = 0.067), with the highest value in the BSFL group and the lowest in BSFP.

The biomarkers of liver and kidney functions are explained in Table 7. The results revealed that all liver function indices (alanine aminotransferase, ALT; aspartate aminotransferase, AST; ALT/AST ratio; and alkaline phosphatase), as well as kidney function biomarkers (urea, Ur; creatinine, Cr; and their ratio, Ur/Cr), did not significantly differ among different protein supplements at 3 or 5%. 

### 3.4. Cecal Microbial Counts

The effects of supplementing different protein sources on cecal microbial counts are delineated in Table 8. The data indicated that clostridial, total coliform, and *Escherichia coli* counts were not discrete among the experimental groups fed FM, BSFL, and BSFP at 3 and 5% during the starter and growing-finishing periods, respectively. Nonetheless, the BSFP groups had significantly lower cecal *Salmonella* and higher *Lactobacillus* spp. counts than the control group and higher total bacterial counts than all the other groups except for BSFL.

### 3.5. Chemical Composition of Excreta

The impact of supplementing disparate protein sources (FM, BSFL, and BSFP) on the chemical composition of the broiler excreta is depicted in Table 9. The data expressed that the dry matter and crude fiber did not significantly vary among the various experimental groups. The FM, BSFL, and BSFP groups voided significantly higher GE than the control groups.

The excreta of the control group only showed higher crude protein than the FM group; the other groups represented intermediate values with differences approaching significance (*p* = 0.069). True protein was significantly lower in the control group than in the BSFP group. The excrement of the BSFP group contained higher levels of EE than the other groups except for BSFL. Also, the BSFL-fed group only voided greater amounts of EE than the control group. The FM group also excreted more EE than the control group. 

## 4. Discussion

The results pinpoint that BSFP contains higher CP (53.5%) than BSFL (50.0%) but lower ether extract. Additionally, both samples had a higher CP content than SBM [28]. The differences in the CP percentages between BSFL and BSFP may be due to the disparate life-cycle stages. These results agree with those reported by Barragan-Fonseca et al. [38], Schiavone et al. [39], Gold et al. [40], and Ahmed et al. [7]. The current results pointed out that BSFL and BSFP reflected comparative nutritive value to SBM and FM [28] when supplemented at 3 and 5% during the starter and growing-finishing periods of broilers. This suggests that both insect products could be adopted as an alternative protein source in poultry nutrition.

The growing performance during days 1–14 was similar among the groups, showing that BSFL and BSFP could be fed to broilers up to 3% in their starter diets without adverse effects on growth rate. Recently, Addeo et al. [11] reported similar results with Japanese quail. 

For 15–42 and 1–42 days of age, the growth performance of broilers that were fed BSFP was decreased by 9% compared to the other groups. These results demonstrated that 5% BSFL could be fed to broiler chickens, yielding a growth rate similar to the control and FM groups, and it did not negatively influence nutrient excretion. The decrease in the growth rate of broilers that were fed BSFP could be attributed to the chitin content, which increases with the insect’s age and, hence, decreases protein digestibility [14,16]. On the other hand, Murawska et al. [41] observed a negative linear relationship between the BW and BWG of broilers that were fed 50%, 75%, and 100% full-fat BSFL meal as a soybean replacement. Also, Facey et al. [42] documented that the body weight of birds exhibited a quadratic decrease when their diets included 12.5% and 25% replacements of SBM with BSFL. Additionally, broiler chickens provided with a diet containing 12.5% BSFL had higher body weights compared to broilers offered diets containing 0%, 50%, and 100% BSFL. However, the decrease in BW was linear with a diet containing 100% BSFL, compared with 0 and 50%, as replacements for SBM. This may be because a diet containing BSFL meal was significantly higher in crude protein and ether extract, so the chickens that were fed this diet had higher final live weights [43]. In addition, Attivi et al. [44] replaced fish meal with BSF larval meal and recognized lower BWG for inclusions of 2, 4, and 6% and higher BWG with an inclusion of 8% that left the total BW unchanged. 

The results of the FCR and EPEI highlight that BSFL can replace SBM and FM at 3 and 5% in starter and grower-finisher diets without a negative effect on the economic traits of broiler chickens. Conversely, the same percentage of BSFP adversely influenced the FCR and PI of broiler chickens. Therefore, the lower growth could have been due to inadequate feed intake. 

The inclusion of BSFL in the starter diet significantly increased the FI by 18.0, 24.2, and 34.3%, compared to the control, FM, and BSFP groups, respectively. The BSFL group also consumed more feed during the growing period, with the increase ranging from 7.46 to 14.3%. For the entire experimental period (1–42 days of age), the feed intake of the BSFL group surpassed that of the other groups by 8.9–16.1%. The increase in the FI of the BSFL group agreed with the results recorded by Dabbou et al. [45], who reported that dietary inclusion of BSFL at 5, 10, and 15% could enhance the feed intake of broilers. Cullere et al. [22] demonstrated that the quail FI did not contrast between animals fed the control diet or the diet supplemented with 10% or 15% defatted BSFL meal. However, Murawska et al. [41] found that broilers that were fed 50%, 75%, and 100% substitutions of SBM with BSFL had negative linear relationships in FI at the end of the experiment. 

The BSFL groups utilized feed less efficiently during the starter period, but the FCR totally recovered during the growing-finishing period, and the whole period showed a similar FCR compared to the control and FM groups. It should be mentioned that the FM groups had the best FCR but did not deviate from the BSFL group. The improved FCR of the FM group was associated with a high growth rate due to better amino acid balance and nutrient digestibility. These results denote that BSFL can be fed to broiler chickens for 1–42 days without adverse effects on feed utilization, and the chemical composition of the manure confirmed this. However, feeding BSFP resulted in worse feed utilization and PI, which may be due to a high chitin content, which can impair growth and feed utilization [14]. Like the present results, Cullere et al. [22] demonstrated that the quail FCRs were not contrasted among young growing quail that were fed a control diet or supplemented with 10% or 15% defatted BSFL meal. In addition, Dabbou et al. [46] found no change in the FCRs of broilers fed diets comprising 5% and 10% BSFL but an impaired FCR in the diet containing 15% compared to the control diet (0%). Along the same line, De Souza Vilela et al. [45] reported that broilers that were fed a full-fat BSFL with up to 20% inclusion had a linear improvement in FCRs during the grower and finisher phases, with no effect on FI. Similar results were published by Mohammed et al. [47], Onsongo et al. [48], and Pieterse et al. [49], who noticed that feeding up to 4% or 15% full-fat BSFP did not impact broiler performance. Nevertheless, Murawska et al. [41] discovered that broilers that were fed 50%, 75%, and 100% substitutions of SBM with BSFL had negatively affected FCRs at the end of the experiment. However, the previous studies point out that the effect of BSFL on feed utilization depends on the poultry species and inclusion level. Also, these results suggest that the protein and amino acid contents of BSFL are as biologically valuable as those of SBM and FM. Our results also concur with Spranghers et al. [21] and Kim et al. [50], who observed that BSFL’s high crude protein and fat content made them an excellent source and valuable component of animal feed nutrition. 

The current results emphasized that almost all blood hematology and biochemical parameters (lipid profiles and their fractions and liver and kidney functions) were not significantly contrasted among the different experimental groups. Similar results were recorded by Van Huis et al. [9] and Fortuoso et al. [51], who noted that BSFs had no effect as a disease vector, high lauric acid (C12:0) formed 64% of the total saturated fatty acid content of the BSFs, and lauric acid was responsible for some of the possible health advantages. Additionally, Londok and Rompis [52] claimed that adding lauric acid to grill diets at 0.03% and 2.6%, respectively, might enhance intestinal health and broiler performance. They also discovered a significant decrease in *E. coli* and other bacteria in excreta samples.

The beneficial effect of BSFP on the bacterial count of broiler chickens agrees with the results of Bernatová et al. [53], who declared that bactericidal antibiotics had a stronger effect than bacteriostatic antibiotics on controlling and suppressing the growth of bacteria. Moreover, supplementation of BSFL and BSFP meals did not cause any allergy side effects for broiler chickens since the lymphocyte transformation test was significantly lower than that recorded for the control and FM groups. The earlier outcomes might be attributed to the insect chitin’s antibacterial properties [54]. Bovera et al. [14] and Khempaka et al. [55] argued that chitin had a beneficial impact on the immune system and, consequently, on the health of chickens, developing their immunological function or decreasing the albumin-to-globulin ratio [23]. 

The prior results reflect the improvement in the cecum microbial colony since the group supplied with BSFP marked the lowest cecum *Salmonella* count. In harmony with the present results, the total count of *Lactobacillus* spp. (a beneficial bacteria) in the cecum of the broilers that were fed a basal diet supplied with BSFP registered the best count. However, the increase in the total bacterial count for the groups fed a basal diet with BSFP may be due to the increased beneficial bacteria recorded for these groups. Accordingly, Park et al. [56] and Diyantor et al. [57] proclaimed that the isolated aqueous extract of the antimicrobial peptides showed broad antibacterial activity. Also, Müller et al. [58] reported that BSFs consisted of antimicrobial peptides like defensins, cecropins, antacids, and diptericins. Similar studies demonstrated that adding BSFL positively impacted broilers’ cecal microbiota and mucin composition [59]. Broilers that were fed BSFL at l28.3 g/kg of DM demonstrated antimicrobial peptides and up to 9% chitin in addition to having high levels of lauric acid, according to De Souza-Vilela et al. [45]. In the same respect, Fortuoso et al. [51] discovered a potential benefit of adding lauric acid to broiler feeds at 0.03% and 2.6%, respectively, on enhancing intestinal health, and both *Escherichia coli* and overall bacterial counts were much lower in excreta samples. In contrast, the fish meal was the only protein source able to reduce the protein content in the excreta in comparison to the control group, suggesting a positive impact on bird production.

Moreover, lauric acid, a kind of dietary monounsaturated fatty acid (MCFA), has a beneficial effect on gut microbiota and has the most potent antibacterial action against *Salmonella*, the bacteria that causes gastroenteritis [60,61]. Moreover, *Lactobacillus* spp. mediates the suppressive effect of MCFA [62]. Based on an in vitro study, gram-positive *Staphylococcus aureus* and gram-negative *Pseudomonas aeruginosa* are just two harmful pathogens that micro-compounds can inhibit from proliferating [56]. The present results suggest that no mortality was recorded during the experimental period. These conclusions are in concert with those of Kawasaki et al. [63], who detected that adding 10% (*w*/*w*) BSF larval meal and 10% (*w*/*w*) prepupae feed to laying hen diets as replacements for soybean meal and oil did not adversely influence the mortality rate.

The current results denote that BSFP can reduce environmental nitrogen pollution in poultry farms due to the high levels of true protein in the excrement. They also exhibited lower gross energy. It was is known that BSFs are an efficient tool for converting organic waste into insect protein and decreasing environmental hazards and organic pollution [64,65].

## 5. Conclusions

It is possible to include BSFL meal as a valuable and sustainable protein source in the starter diet of broilers at 3% and in their growing-finishing diet at 5% at 1–14 and 15–42 days of age, respectively, without adverse effects on growth performance and environmental sustainability. BSFL meal yielded comparable feed utilization, growth rates, and production indexes to SBM and FM. In addition, the inclusion of BSLP meal in broiler diets decreased *Salmonella*, while increasing Lactobacillus spp. and total bacterial counts. Moreover, the hematochemical, liver, and renal functions and environmental sustainability were conserved by dietary BSFL and BSFP inclusion. This signifies that BSFL are a potential novel feed ingredient for growing broiler chickens during 1–42 days of age, and the effects of BSFP on gut microbials need further research.

## Figures and Tables

**Table 1 animals-13-02759-t001:** Chemical analysis of the experimental ingredients exploited in the diet formulation.

	Yellow Corn	Soybean Meal	Fish Meal	BSF Larvae	BSF Prepupae
Dry matter, %	91.0	89.0	91.1	90.2	90.8
Organic matter, %	87.3	83.8	86.3	87.2	88.3
Gross energy, kcal/kg	3943	4200	4545	4428	4450
Crude protein, % DM	8.50	46.20	60.04	50.00	53.5
Ether extract, % DM	6.50	3.00	9.69	22.0	20.0
Crude fiber, % DM	5.80	6.10	1.15	7.40	7.00
Crude ash, % DM	3.70	5.20	4.84	3.00	2.50
NFE, % DM	66.5	28.5	15.4	7.75	7.79
Chitin, % DM	--	--	--	4.92	5.17
Calcium, % DM	0.03	0.039	0.496	0.295	0.28
Phosphorus, % DM	0.087	0.066	0.219	0.505	0.202

DM: dry matter; NFE: nitrogen-free extracts; BSF: black soldier fly.

**Table 2 animals-13-02759-t002:** Ingredients and chemical composition (g/kg) of the experimental diets.

	Starter Diets, g/kg				Growing-Finishing Diets, g/kg			
	Control	FM	BSFL	BSFP	Control	FM	BSFL	BSFP
Ingredients								
Yellow corn	570.3	583.3	566.6	564.6	617.0	656.0	628.4	628.0
Soybean meal	355	323	335	335	305	240	255	255
Fish meal	0.0	30.0	0.0	0.0	0.0	50.0	0.0	0.0
BSFL	0.0	0.0	30.0	0.0	0.0	0.0	50.0	0.0
BSPP	0.0	0.0	0.0	30.0	0.0	0.0	0.0	50.0
Vit + Min Premix ^1^	2.00	2.00	2.00	2.00	2.00	2.00	2.00	2.00
NaCl	4.00	4.00	4.00	4.00	4.00	4.00	4.00	4.00
Ca(H_2_PO_4_)	21.7	20.0	17.0	20.0	15.0	11.0	5.00	11.0
CaCO_3_	9.00	8.00	11.0	10.0	7.00	4.00	12.0	6.4
Soybean oil	33.5	25.5	30.0	30.0	46.0	30.0	39.0	39.0
Lysine	2.00	2.00	2.00	2.00	2.00	1.40	2.60	2.60
Dl-methionine	2.50	2.15	2.40	2.40	2.00	1.60	2.00	2.00
Calculated composition, g/kg								
Calcium	12.1	12.6	12.2	12.7	9.0	8.9	9.0	8.8
Av. phosphorus	6.08	6.29	6.52	6.26	4.54	4.63	4.79	4.57
Lysine	12.5	13.0	12.5	12.5	11.3	11.4	11.4	11.4
Methionine	5.68	5.65	5.61	5.69	4.96	5.05	4.94	5.09
Methionine + cysteine (TSAA, %)	9.05	9.00	9.03	9.04	8.09	8.10	8.07	8.11
Determined composition, g/kg								
Crude protein	218.8	222.0	223.3	225.5	196.1	193.6	193.7	201.5
AME, kcal/kg	2921	2901	2850	2880	3085	3125	3075	3069
Ether extracts	56.5	54.1	57.5	59.3	61.6	66.1	66.0	65.9
Crude fiber	43.3	44.4	42.4	44.6	44.4	44.2	46.3	45.9
Ash, %	11.3	11.2	11.2	11.1	10.9	11.0	10.0	10.8
Dry matter, %	904.3	901.4	899.8	89.55	900.6	903.1	898.0	895.2

^1^ Super-mix vitamin and trace elements of premix antioxidant were added per 2 kg. Each 1 kg contained 6250 mg of vitamin A, 2500 mg of vitamin D3, 250 mg of vitamin E, 5 mg of vitamin B1, 27.5 mg of vitamin B2, 12.5 mg of vitamin B6, 10 μg of vitamin B12, 17.5 mg of vitamin K3, 20 mg of nicotinic acid, 50 mg of calcium pantothenate, 5 mg of folic acid, 50 μg of biotin, 220 mg Fe as iron-II-sulfate, 25 mg Cu as copper-II-sulfate, 21 mg Mn as manganese-II-oxide, 375 mg Zn as zinc oxide, 6.5 mg I as potassium iodate, 1.13 mg Se as sodium selenite, 1.13 mg Co as cobalt-sulfate, 2.5 mg of ethoxyquin, choline colored tars, 120 g of wheat bran, and limestone.

**Table 3 animals-13-02759-t003:** Effect of dissimilar dietary supplementary protein sources on growth of broiler chickens from 1 to 42 days of age.

Treatments	Initial BW g	BWG 1–14 Days g	BWG 15–42 Days g	BWG 1–42 Days g	BW 42 Days g
Control	42.1	292	1800 ^a^	2092 ^a^	2134 ^a^
Fish meal	42.8	292	1768 ^a^	2060 ^a^	2103 ^a^
BSFL	43.1	282	1798 ^a^	2080 ^a^	2123 ^a^
BSFP	43.2	299	1425 ^b^	1724 ^b^	1767 ^b^
SEM	2.45	11.4	36.1	62.8	62.8
*p*-value	0.857	0.573	0.0004	0.002	0.002

^a,b^: *p* < 0.05; BW: body weight; BWG: body weight gain; BSFL: black soldier fly larvae; BSFP: black soldier fly prepupae; SEM: standard error of the mean; n = 6.

**Table 4 animals-13-02759-t004:** Effects of divergent dietary supplementary protein sources on feed intake and feed conversion.

Treatments	FI 1–14 Days g/bird	FI 15–42 Days g/bird	FI 1–42 Days g/bird	FCR 1–14 Days kg/kg	FCR 15–42 Days kg/kg	FCR 1–42 Days kg/kg	EPEI
Control	561 ^b^	2866 ^ab^	3427 ^b^	1.94 ^b^	1.59 ^c^	1.83 ^bc^	273 ^a^
Fish meal	533 ^bc^	3000 ^a^	3533 ^b^	1.93 ^b^	1.70 ^bc^	1.71 ^c^	288 ^a^
BSFL	662 ^a^	3224 ^a^	3886 ^a^	2.36 ^a^	1.79 ^abc^	2.01 ^ab^	246 ^a^
BSFP	493 ^c^	2820 ^b^	3313 ^b^	1.66 ^b^	1.97 ^a^	2.21 ^a^	188 ^b^
SEM	18.32	40.56	75.99	0.123	0.054	0.071	11.06
*p*-value	0.0001	0.002	0.0003	0.001	0.001	0.0001	0.0001

^a,b,c^: *p* < 0.05; FI: feed intake; FCR: feed conversion ratio; EPEI: European Production Efficiency Index; BSFL: black soldier fly larvae; BSFP: black soldier fly prepupae; SEM: standard error of the mean; n = 6.

**Table 5 animals-13-02759-t005:** Effects of discrete dietary supplementary protein sources fed from 1 to 42 days of age on red blood cell characteristics of broiler chickens at 42 days of age.

Treatments	RBC 10^6^/cm	Hgb %	PCV %	MCV µm^3^/RBC	MCH pg/dL	MCHC %
Control	2.03	12.38	36.54	182	61.69	33.92
Fish meal	1.89	12.10	36.74	194	64.18	32.92
BSFL	1.81	11.04	33.05	188	62.80	33.37
BSFP	1.82	12.03	36.87	203	66.27	32.65
SEM	0.716	0.415	1.219	12.205	3.76	0.531
*p*-value	0.085	0.616	0.059	0.115	0.086	0.494

RBC: red blood cells; Hgb: hemoglobin; PCV: packed cell volume; MCV: mean corpuscular volume; MCH: mean corpuscular hemoglobin; MCHC: mean corpuscular hemoglobin concentration; BSFL: black soldier fly larvae; BSFP: black soldier fly prepupae; SEM: standard error of the mean; n = 6.

**Table 6 animals-13-02759-t006:** Effects of divergent dietary supplementary protein sources fed from 1 to 42 days of age on lipid profiles of broiler chickens at 42 days of age.

Treatments	TL mg/dL	Trgs mg/dL	Chol mg/dL	RHC	HDL mg/dL	LDL mg/dL	HDL/LDL	VLDL mg/dL
Control	606	178	207	0.429	40.2	88.9	0.454	35.6
Fish meal	610	174	203	0.459	42.3	93.1	0.456	34.8
BSFL	556	180	203	0.464	42.3	94.0	0.452	36.0
BSFP	545	180	214	0.414	39.9	88.7	0.423	36.0
SEM	3.94	3.162	3.955	0.013	1.064	2.150	0.018	0.633
*p*-value	0.488	0.667	0.212	0.067	0.224	0.260	0.676	0.667

TL: total lipids; Trgs: triglycerides; Chol: cholesterol; RHC: risk of high cholesterol; HDL: high-density lipoprotein; LDL: low-density lipoprotein; VLDL: very low-density lipoprotein; BSFL: black soldier fly larvae; BSFP: black soldier fly prepupae; SEM: standard error of the mean; n = 6.

**Table 7 animals-13-02759-t007:** Effects of different dietary supplementary protein sources fed from 1 to 42 days of age on liver and kidney indices of broiler chickens at 42 days of age.

Treatments	ALT IU/L	AST IU/L	ALT/AST	ALP u/100 mL	Urea mg/dL	Cr mg/dL	Urea/Cr
Control	67.98	58.20	1.17	75.28	29.98	29.04	1.03
Fish meal	66.61	57.38	1.17	76.82	29.10	28.05	1.04
BSFL	66.68	58.06	1.17	76.26	27.93	27.81	1.00
BSFP	65.71	55.26	1.19	75.68	30.41	28.54	1.08
SEM	0.935	1.358	0.037	1.174	1.188	1.259	0.052
*p*-value	0.557	0.543	0.910	0.331	0.095	0.084	0.094

ALT: alanine aminotransferase; AST: aspartate aminotransferase; ALP: alkaline phosphates; Cr: creatinine; BSFL: black soldier fly larvae; BSFP: black soldier fly prepupae; SEM: standard error of the mean; n = 6.

**Table 8 animals-13-02759-t008:** Effects of contrasting dietary supplementary protein sources fed from 1 to 42 days of age on cecum microbial counts of broiler chickens at 42 days of age.

	*Clostridia* Cfu/ml	*Salmonella* Cfu/mL	TC Cfu/mL	*E. coli* Cfu/mL	*Lactobacillus* Cfu/mL	TBC Cfu/mL
Control	1.09	3.10 ^ab^	3.78	3.09	5.23 ^b^	5.84 ^b^
Fish meal	1.45	3.22 ^a^	3.90	2.72	5.96 ^ab^	6.55 ^b^
BSFL	1.38	2.92 ^ab^	4.30	3.33	6.29 ^ab^	7.55 ^ab^
BSFP	2.03	2.39 ^b^	4.30	2.83	6.69 ^a^	9.04 ^a^
SEM	0.749	0.264	0.164	0.217	0.317	0.417
*p*-value	0.513	0.008	0.107	0.261	0.045	0.001

^a,b^: *p* < 0.05; TC: total coliforms; TBC: total bacterial count; BSFL: black soldier fly larvae; BSFP: black soldier fly prepupae; SEM: standard error of the mean; n = 6.

**Table 9 animals-13-02759-t009:** Effect of the diets on excreta analyses of broiler chickens at 42 days of age.

Treatments	DM %	GE kcal/kg	CP %	TP %	EE %	CF %	Ash %
Control	27.8	1446 ^b^	23.3 ^a^	5.23 ^b^	4.61 ^c^	10.4	26.7
Fish meal	28.1	1525 ^a^	22.7 ^b^	5.96 ^ab^	4.65 ^bc^	10.3	26.7
BSFL	27.6	1347 ^c^	23.1 ^ab^	6.29 ^ab^	4.68 ^ab^	10.3	26.7
BSFP	27.9	1347 ^c^	23.0 ^ab^	6.69 ^a^	4.71 ^a^	10.4	26.8
SEM	0.211	1.83	0.136	0.021	0.038	0.053	0.4341
*p*-value	0.304	0.0001	0.069	0.041	0.0001	0.456	0.883

^a,b,c^: *p* < 0.05; DM: dry matter; GE: gross energy; CP: crude protein; TP: true protein; EE: ether extract; CF: crude fiber; BSFL: black soldier fly larvae; BSFP: black soldier fly prepupae; SEM: standard error of the mean; n = 6.

## Data Availability

Please direct any data requests to the corresponding authors of the study.

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
