# Peer review of "Fish and Black Soldier Fly Meals as Partial Replacements for Soybean Meal Can Affect Sustainability of Productive Performance, Blood Constituents, Gut Microbiota, and Nutrient Excretion of Broiler Chickens"

_animals, 2023, doi:10.3390/ani13172759_

Round 1
Reviewer 1 Report
The topic is very accurate and interesting, however the article contains some methodological issues, which indicates it must be rewritten.
In the materials and methods chapter, there is only 120 animals in the total study, which is number is quite low for a performance study. I suggest to delete the first part of the results, focusing only the blood parameters and the excreta.
I am also concerned about the housing management of the birds (cage and cage size). Was a university licence enough for this experiment?
In this respect, I suggest to re-organize the manuscript with a major revision.
-
Author Response
The topic is very accurate and interesting; however the article contains some methodological issues, which indicates it must be rewritten.
Au: thank you very much for your encouraging comment. We applied all the required changes.
In the materials and methods chapter, there is only 120 animals in the total study, which is number is quite low for a performance study.
Au: dear reviewer, looking at other manuscript published in the topic, the number of animals can be considered adequate for the study. In addition, the number of animals is important, but in research involving poultry, a very important aspect is the number of replicates. In our trial, we have 6 replicates each of 5 birds. This can be considered enough using estimated sample size by SAS, stated in the statistical section.
I suggest to delete the first part of the results, focusing only the blood parameters and the excreta.
Au: Thank you for your suggestion although as indicated in above section, the number of animals/replicates is adequate and growth performance is essential part of this type of the work. Moreover, the other 4 reviewers did not raise you issue.
I am also concerned about the housing management of the birds (cage and cage size). Was a university license enough for this experiment?
Au: dear reviewer, thank you for your concern for animal welfare, however, the surface area of each cage was 80 x 50 = 4,000 cm2. Each cage is for one replicate and each replicate contains 5 birds. So, each bird has 800 cm2 with a heigh of 45 cm. We understand that cage farming of broiler is not a common practice in EU, but it is common for chickens’ experiment. The available space for each bird was higher than the minimum required by European Union for laying hens (750 cm2)
Reviewer 2 Report
The manuscript entitled "Black soldier fly larvae and prepupae affect sustainability of 2 productive performance, blood constituents, gut microbiota 3 and nutrients excretion of broiler chickens" was well presented and written. This manuscript provides scientific evidence of an alternative protein source from insects to replace soybean or fish meal in animal diets. In my opinion, the manuscript is appropriate to publish in Animals with minor correction based on my comments below:
1- The author should use an appropriate title that is consistent with the Table captions in the results section. In the result section, the author focuses on the effect of protein sources instead of black soldier fly larvae and prepupae as described in the title.
2. Regarding BSFL and BSFP preparation. The authors mentioned that they were bought from the local market, so how did the authors control the quality and familiarity of these ingredients?
Authors have to describe the details about the ways to dry and ground the insects, as well as how to include them in the diets.
3- Table 3: SEM and P value of survival rate
4- If the authors would look at the effects of protein sources, the data of BSFL and BSFP should be pooled and analyzed the protein source were soybean, fish meal and insect protein.
5- Authors should present the number after decimal consistently in all tables
Author Response
The manuscript entitled "Black soldier fly larvae and prepupae affect sustainability of productive performance, blood constituents, gut microbiota and nutrients excretion of broiler chickens" was well presented and written. This manuscript provides scientific evidence of an alternative protein source from insects to replace soybean or fish meal in animal diets. In my opinion, the manuscript is appropriate to publish in Animals with minor correction based on my comments below:
Au: thank you very much for your valuable comment.
1- The author should use an appropriate title that is consistent with the Table captions in the results section. In the result section, the author focuses on the effect of protein sources instead of black soldier fly larvae and prepupae as described in the title.
Au: you are right, the title of the manuscript has been changed to “Fish and black soldier fly meals as partial replacement of soybean meal can affect sustainability of productive performance, blood constituents, gut microbiota and nutrients excretion of broiler chickens”
- Regarding BSFL and BSFP preparation. The authors mentioned that they were bought from the local market, so how did the authors control the quality and familiarity of these ingredients?
The quality was control by investigating the general and physical appearance and chemical composition of the BSFL and BSFP.
Authors have to describe the details about the ways to dry and ground the insects, as well as how to include them in the diets.
Au: BSFL and BSFP were dried in forced hot air oven at 65 ºC until constant weight and grounded using an electric dry mill to fine powder of 0.5 mm sieve. The BSFL and BSFP were included in the were included in in the formulation of the experimental diets biweekly and mixed well for ten minutes until the diets were totally homogenized.
3- Table 3: SEM and P value of survival rate
Au: survival rate was non statistically analyzed. The P value and SEM were deleted from the table.
4- If the authors would look at the effects of protein sources, the data of BSFL and BSFP should be pooled and analyzed the protein source were soybean, fish meal and insect protein.
Au: you are right, the title and the aim of the paper have been changed
5- Authors should present the number after decimal consistently in all table
Au: we done our best to normalize the tables, however, the use of number and decimals are constant with the international standard: xxxx, xxx, xx.x, x.xx, 0.xxx and 0.0xxx
Reviewer 3 Report
Please review word usage in introduction and conclusion. Review tables and spacing within the columns and column headings.
In MM you mention weighing at 28 days but no data in the table.
In results, you mention day 29-42 but there is no data for that time period in the table.
100 % survivability? - no birds removed throughout the study?
NA
Author Response
Please review word usage in introduction and conclusion. Review tables and spacing within the columns and column headings.
Thank you for your advice, we revised the Ms, and the Ms will be will further checked by the editor office in GP stage.
In MM you mention weighing at 28 days but no data in the table.
Au: you are right there was a mistake. It is not 28 but 14, corrected.
In results, you mention day 29-42 but there is no data for that time period in the table.
Au: the same mistake, thank you very much, corrected
100 % survivability? - no birds removed throughout the study?
Au: exactly. In this trial we had no mortality
Author Response
All the required changes have been done
The CRD was used for the experimental design and sample size was determined using the SAS program:
All the questions about statistical analysis were answered in the M&M section.
it will be more suitable if authors evaluate orthogonal polynomial contrast for all the parameters. The one-way ANOVA and Tukey post-hock give similar results to orthogonal polynomial contrast and more applicable results for the end user of broiler farming application.
The 4 analysis of variance assumptions (ANOVA) Inculded: 1) Indivdual observation are mutually independent, 2) the data adhere to an additive effect statisitcal model comprsing fixed effects and random errors, 3) the random error are normally distributed and 4) the random errors have homogenous variantion.
What about the random effect was tested, see item 3 and 4 of the above section.
Reviewer 5 Report
Dear Authors,
I have revised the manuscript entitled “Black soldier fly larvae and prepupae affect sustainability of 2 productive performance, blood constituents, gut microbiota 3 and nutrients excretion of broiler chickens”. This study investigated the impact of black soldier fly larvae (BSFL), and prepupae (BSFP) meals added to feed as partial replacement of soybean meal, on growth performance, blood constituents, and gut microbiota of broiler chickens in comparison to the control diets containing soybean meal and/or fish meal as main protein sources.The methodology used is rather classic (not modern) for this type of analysis and is often used in mammal and birds` models.The experiment was carried out correctly, and subsequent analyzes seem to be consistent and logical. The obtained results are presented correctly, they allow to draw the conclusions presented by the authors of the study. The presented study shows a standard research level, consistent with current observations in this thematic area. The paper is interesting, but I have some other comments. Please, see below:
First of all, please check the English language, there are some stylish errors, missed interpunction marks, repetition of similar/the same words. Some sentences are to complex and need to be cut for simple, short sentences (for easier understanding- e.g., the abstract)
L-83- Did the insects used have quality certificates? have been standardized according to the rules applicable to classic feed ingredients and feed additives? Whether the raw material and the product had microbiological tests - this is particularly important in the tests of the immune system assessment. How can be sure that the raw material was standardised?
L-107- does the feed was divided into grower 1 and grower 2? Please, give the age of the birds for each type of feed : G1-from….to, G2 from… to.. finisher from.. to..
Was any coccidiostats use in the feed?
What was the vaccination program?
Were any extra supplements or acidifier use in drinking water?
Table 1
Please correct the format and the size of the letters : Soybea n meal
unify the names of ingredients - crude protein or Crude Protein, Crude Ash, Crude Fiber, etc.
Did you check and compare the size of the critical organs (kidney, liver) and link the results with those from blood? Did you make a histological analysis of the liver and kidney tissues to check if there is nothing there that could indicate the development of pathological changes associated with the use of these two additives?
The high costs of classic protein raw materials in broiler chicken feed have been raised many times in the text. Do you have information about the price of the raw material used in the study? According to your experience and current literature data, what should be the amounts (in percent or g/kg per ton of classic feed) of black soldier additives (different forms) in order to obtain the best production parameters, and at the same time to make the price of this type of feed at an acceptable level for broiler breeders.
In the majority of studies analyzing the composition of the intestinal flora, classical cultures are abandoned due to the limitations of this method (media), more and more often molecular biology methods are becoming routine - more sensitive, faster, allowing for extensive analyzes in a short time. The most commonly used are NGS methodology, why did you choose classic methods for yours analyzes and not NGS?
There are some stylish errors, missed interpunction marks, repetition of similar/the same words. Some sentences are to complex and need to be cut for simple, short sentences (for easier understanding- e.g., the abstract)
Please, revised the English with the native speaker.
Author Response
I have revised the manuscript entitled “Black soldier fly larvae and prepupae affect sustainability of 2 productive performance, blood constituents, gut microbiota 3 and nutrients excretion of broiler chickens”. This study investigated the impact of black soldier fly larvae (BSFL), and prepupae (BSFP) meals added to feed as partial replacement of soybean meal, on growth performance, blood constituents, and gut microbiota of broiler chickens in comparison to the control diets containing soybean meal and/or fish meal as main protein sources.The methodology used is rather classic (not modern) for this type of analysis and is often used in mammal and birds` models.The experiment was carried out correctly, and subsequent analyzes seem to be consistent and logical. The obtained results are presented correctly, they allow to draw the conclusions presented by the authors of the study. The presented study shows a standard research level, consistent with current observations in this thematic area. The paper is interesting, but I have some other comments. Please, see below:
Au: thank you for your considerations
First of all, please check the English language, there are some stylish errors, missed interpunction marks, repetition of similar/the same words. Some sentences are to complex and need to be cut for simple, short sentences (for easier understanding- e.g., the abstract)
Au: the English has been reviewed
L-83- Did the insects used have quality certificates? have been standardized according to the rules applicable to classic feed ingredients and feed additives? Whether the raw material and the product had microbiological tests - this is particularly important in the tests of the immune system assessment. How can be sure that the raw material was standardised?
Au: BSFL and BSFP were purchased from the local market of commercial supply and chemical analyzed to confirm the standard publish data for insects. The insects were checked for general appearance. The insects were in good form as no sign of diarrhea and salmonella were observed in chickens after feeding and as also confirmed by absent of mortality and normal gut microbiota.
L-107- does the feed was divided into grower 1 and grower 2? Please, give the age of the birds for each type of feed : G1-from….to, G2 from… to.. finisher from.. to..
Au: the specification has been added
Was any coccidiostats use in the feed?
Au: No coccidiostats or antibiotics were applied in this research work.
What was the vaccination Non program?
Au: No vaccination was applied in this research work as broilers were clean environment house that was used for the 1st time.
Were any extra supplements or acidifier use in drinking water?
Au: No extra supplements or acidifier were applied in this research work.
Table 1
Please correct the format and the size of the letters : Soybea n meal
Au: corrected
unify the names of ingredients - crude protein or Crude Protein, Crude Ash, Crude Fiber, etc.
Au: corrected
Did you check and compare the size of the critical organs (kidney, liver) and link the results with those from blood? Did you make a histological analysis of the liver and kidney tissues to check if there is nothing there that could indicate the development of pathological changes associated with the use of these two additives?
Au: We have looked at the kidney, liver, spleen and gut organs and were in normal signs when compared with the literature values and as confirmed by the histopathology and morphometry study, however, we will published the data or carcass traits, meat quality and body organs and histopathology and morphometry, and digestibility in further article.
The high costs of classic protein raw materials in broiler chicken feed have been raised many times in the text. Do you have information about the price of the raw material used in the study? According to your experience and current literature data, what should be the amounts (in percent or g/kg per ton of classic feed) of black soldier additives (different forms) to obtain the best production parameters, and at the same time to make the price of this type of feed at an acceptable level for broiler breeders.
Au: dear reviewer, actually the production cost of insect meals is in general, high…higher than fishmeal….even if there a decreasing trend. We need a massive industrial production to obtain an effective reduction in the insect meals price. Regarding the inclusion level in the diets: according to our experience the best choice is to add low percentages of insect meals. In fact high inclusion levels had a controversial effects due to the presence of chitin.
In the majority of studies analyzing the composition of the intestinal flora, classical cultures are abandoned due to the limitations of this method (media), more and more often molecular biology methods are becoming routine - more sensitive, faster, allowing for extensive analyzes in a short time. The most commonly used are NGS methodology, why did you choose classic methods for yours analyzes and not NGS?
Au: You are absolute right and we agree with you, but we suffer from adequate fund NGS, we have been wished to do this technical and trustful method, any way the used methods was done by expertise and gave acceptable results.
Comments on the Quality of English Language
There are some stylish errors, missed interpunction marks, repetition of similar/the same words. Some sentences are to complex and need to be cut for simple, short sentences (for easier understanding- e.g., the abstract)
Please, revised the English with the native speaker.
Au: the English has been revised
Round 2
Reviewer 5 Report
The new version of the manuscript, with the corrections made, looks much better. In my opinion, the study in this form qualifies for publication.
Author Response
.